# BAYESADAPTER: BEING BAYESIAN, INEXPENSIVELY AND ROBUSTLY, VIA BAYESIAN FINE-TUNING

## ABSTRACT

Despite their theoretical appealingness, Bayesian neural networks (BNNs) are falling far behind in terms of adoption in real-world applications compared with deterministic NNs, mainly due to their limited scalability in training and low fidelity in uncertainty estimates. In this work, we develop a new framework, named *BayesAdapter*, to address these issues and bring Bayesian deep learning to the masses. The core notion of *BayesAdapter* is to adapt pre-trained deterministic NNs to be BNNs via *Bayesian fine-tuning*. We implement *Bayesian fine-tuning* with a plug-and-play instantiation of stochastic variational inference, and propose *exemplar reparameterization* to reduce gradient variance and stabilize the fine-tuning. Together, they enable training BNNs as if one were training deterministic NNs with minimal added overheads. During *Bayesian fine-tuning*, we further propose an uncertainty regularization to supervise and calibrate the uncertainty quantification of learned BNNs at low cost. To empirically evaluate *BayesAdapter*, we conduct extensive experiments on a diverse set of challenging benchmarks, and observe satisfactory training efficiency, competitive predictive performance, and calibrated and faithful uncertainty estimates.

## 1 INTRODUCTION

Much effort has been devoted to developing flexible and efficient Bayesian deep models to make accurate, robust, and well-calibrated decisions (MacKay, 1992; Neal, 1995; Graves, 2011; Blundell et al., 2015), with Bayesian neural networks (BNNs) as popular examples. The principled uncertainty quantification inside BNNs is critical for realistic decision-making, well evaluated in scenarios ranging from model-based reinforcement learning (Depeweg et al., 2016) and active learning (Hernández-Lobato & Adams, 2015), to healthcare (Leibig et al., 2017) and autonomous driving (Kendall & Gal, 2017). BNNs are also known to be capable of resisting over-fitting.

However, there are fundamental obstacles posed in front of ML practitioners when trying to push the limit of BNNs to larger datasets and deeper architectures: (*i*) The scalability of the existing BNNs is generally restrictive owing to the essential difficulties of learning a complex, non-degenerate distribution over parameters in a high-dimensional and over-parameterized space (Liu & Wang, 2016; Louizos & Welling, 2017; Sun et al., 2019). (*ii*) The Bayes posteriors learned from scratch are often systematically worse than their point-estimate counterparts in terms of predictive performance when "cold posterior" strategies are not applied (Wenzel et al., 2020). (*iii*) It is shown that the BNNs have the possibility to assign low (*epistemic*) uncertainty for realistic out-of-distribution (OOD) data (e.g., adversarial examples), rendering their uncertainty estimates unreliable in safety-critical scenarios (Grosse et al., 2018).

To solve these problems, we present a scalable workflow, named *BayesAdapter*, to learn more reliable BNNs. In a holistic view, we unfold the learning of a BNN into two steps: *deterministic pre-training* of the deep neural network (DNN) counterpart of the BNN followed by several-round *Bayesian fine-tuning*. This enables us to learn a principled BNN with slightly more efforts than training a regular DNN, and provides us with the opportunities to embrace qualified off-the-shelf pre-trained DNNs (e.g., those on PyTorch Hub). The converged parameters of the deterministic model serve as a strong start point for *Bayesian fine-tuning*, allowing us to bypass extensive local

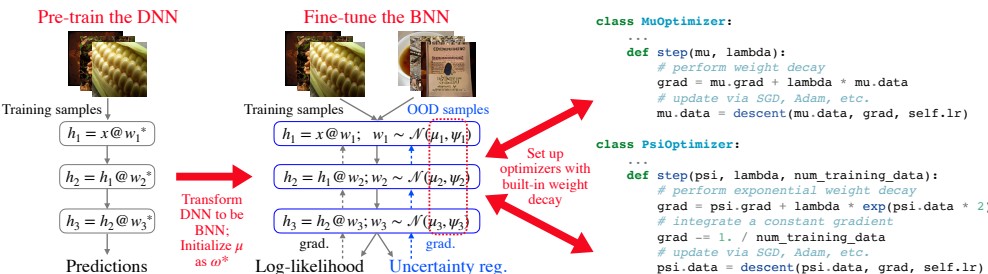

**Figure 1:** The workflow of *BayesAdapter*. We assume a three-layer model for simplicity. We at first pre-train a DNN counterpart of the target BNN via *maximum a posteriori* (MAP) estimation, then transform it to be a BNN by replacing the point-estimate parameters with a diagonal Gaussian centered at them, from which the parameter samples are drawn for computation. After that, we build separate optimizers with built-in weight decay for the Gaussian mean and variance, and perform fine-tuning to fit the data under uncertainty regularization based on autodiff libraries.

optimum suffered by a direct learning of BNN[1]. To render the fine-tuning in the style of training normal NNs, we resort to stochastic variational inference (VI) to update the approximate posterior. We develop optimizers with built-in weight decay for the parameters of the variational distribution to absorb the regularization effects from the prior, and develop *exemplar reparametrization* to reduce the gradient variance. Moreover, to make the uncertainty estimation of the learned models reliable, we propose to additionally, explicitly regularize the model to behave uncertainly on representative foreseeable OOD data during fine-tuning. This regularization takes the form of a margin loss, and is readily applicable to most of the existing BNNs. Figure 1 depicts the whole framework of *BayesAdapter*. Extensive empirical studies validate the efficiency and effectiveness of our workflow. In summary, our contributions are as follows:

1. We propose *BayesAdapter*, to quickly and cheaply adapt a pre-trained DNN to be Bayesian without compromising performance when facing new tasks.

2. We provide an easy-to-use instantiation of stochastic VI, which allows learning a BNN as if training a deterministic NN and frees the users from tedious details of BNN.

3. We augment the fine-tuning with a generally applicable uncertainty regularization term to rectify the predictive uncertainty according to a collection of OOD data.

4. Extensive studies validate that BayesAdapter is scalable; the delivered BNN models are high-quality; and the acquired uncertainty quantification is calibrated and transferable.

## 2 BAYESADAPTER

In this section, we first motivate *BayesAdapter* by drawing a connection between *maximum a posteriori* (MAP) and Bayesian inference. We then describe the proposed procedure *Bayesian fine-tuning*, and a practical and robust implementation of stochastic VI to realize it. Figure 1 illustrates the overall workflow of *BayesAdapter*.

### 2.1 FROM DNNS TO BNNS

Let $\mathcal{D} = \{(\boldsymbol{x}_i, y_i)\}_{i=1}^n$ be a given training set, where $\boldsymbol{x}_i \in \mathbb{R}^d$ and $y_i \in \mathcal{Y}$ denote the input data and label, respectively. A DNN model can be fit via MAP as following:

$$\max_{\boldsymbol{w}} \frac{1}{n} \sum_i [\log p(y_i|\boldsymbol{x}_i; \boldsymbol{w})] + \frac{1}{n} \log p(\boldsymbol{w}). \tag{1}$$

We use $\boldsymbol{w} \in \mathbb{R}^p$ to denote the high-dimensional model parameters, and $p(y|\boldsymbol{x}; \boldsymbol{w})$ as the predictive distribution associated with the model. The prior term $p(\boldsymbol{w})$, when taking the form of an isotropic Gaussian, reduces to the common L2 weight decay regularizer in optimization. Despite the wide adoption, DNNs are known to be prone to over-fitting, generating over-confident predictions, and are unable to convey valuable information on the trustworthiness of their predictions. Naturally, Bayesian neural networks (BNNs) come into the picture to address these limitations.

---

[1]Here the BNN mainly refers to mean-field variational BNNs, and the results in Sec 4.1 testify this point.

Typically, a BNN imposes a prior $p(\boldsymbol{w})$ on model parameters, which is put together with the likelihood $p(\mathcal{D}|\boldsymbol{w})$ to infer the posterior $p(\boldsymbol{w}|\mathcal{D})$. Among the wide spectrum of BNN algorithms (MacKay, 1992; Neal, 1995; Graves, 2011; Blundell et al., 2015; Liu & Wang, 2016; Gal & Ghahramani, 2016; Louizos & Welling, 2017), variational BNNs are particularly promising due to their ease of training compared with other BNN variants. Formally, variational BNNs derive a $\boldsymbol{\theta}$-parameterized varitional distribution $q(\boldsymbol{w}|\boldsymbol{\theta})$ to approximate the true posterior $p(\boldsymbol{w}|\mathcal{D})$, by maximizing the evidence lower bound (ELBO) (scaled by $1/n$):

$$\max_{\boldsymbol{\theta}} \underbrace{\mathbb{E}_{q(\boldsymbol{w}|\boldsymbol{\theta})}\left[\frac{1}{n}\sum_i \log p(y_i|\boldsymbol{x}_i;\boldsymbol{w})\right]}_{\mathcal{L}_{ell}} - \underbrace{\frac{1}{n}D_{\mathrm{KL}}\left(q(\boldsymbol{w}|\boldsymbol{\theta})\|p(\boldsymbol{w})\right)}_{\mathcal{L}_c}, \tag{2}$$

where $\mathcal{L}_{\mathrm{ell}}$ is the *expected log-likelihood* and $\mathcal{L}_c$ is the *complexity loss*. By casting posterior inference into optimization, Eq. (2) makes the training of BNNs more approachable. However, most existing BNNs[2] trained under such a criterion exhibit limitations in scalability and performance (Osawa et al., 2019a; Wenzel et al., 2020) compared with their deterministic counterparts, mainly attributed to the higher difficulty of learning high-dimensional distributions than point estimates, and challenges in finding non-degenerated optima of highly nonlinear functions characterized by NNs.

Given that MAP converges to the mode of the Bayesian posterior, it might be plausible to adapt pre-trained deterministic DNNs to be Bayesian economically. Following this hypothesis, we propose to repurpose the converged parameters $\boldsymbol{w}^*$ of MAP, and use it to instantiate $q(\boldsymbol{w}|\boldsymbol{\theta})$ as a Gaussian $\mathcal{N}(\boldsymbol{w};\boldsymbol{\theta})$ with $\boldsymbol{\theta} = (\boldsymbol{\mu},\boldsymbol{\Sigma})$, where $\boldsymbol{\mu}$ is initialized as $\boldsymbol{w}^*$ and $\boldsymbol{\Sigma} \in \mathbb{R}^{p \times p}$ denotes the covariance. Then, we arrive at a BNN with *posterior predictive*:

$$p(y|\boldsymbol{x},\mathcal{D}) = \mathbb{E}_{\mathcal{N}(\boldsymbol{w};\boldsymbol{\mu},\boldsymbol{\Sigma})}p(y|\boldsymbol{x};\boldsymbol{w}) \approx \frac{1}{S}\sum_{s=1}^{S} p(y|\boldsymbol{x};\boldsymbol{w}^{(s)}), \text{where } \boldsymbol{w}^{(s)} \sim \mathcal{N}(\boldsymbol{w};\boldsymbol{\mu},\boldsymbol{\Sigma}), s = 1,...,S. \tag{3}$$

Eq. (3) is also called *Bayes ensemble*, where $\boldsymbol{\mu}$ is perturbed, and the predictions from multiple likely models are assembled. $\boldsymbol{\Sigma}$ controls the magnitude of perturbation. A classic method to generate an informative $\boldsymbol{\Sigma}$ is by Laplace approximation (Bleistein & Handelsman, 1986), but it is more like a postprocessing procedure, lacking the flexibility to jointly adapt the mean and covariance of the Gaussian posterior w.r.t. data, and its naive implementation without strong assumptions may be computationally prohibitive. Instead, we suggest a more practical workflow – that fine-tunes the approximate posterior $\mathcal{N}(\boldsymbol{w};\boldsymbol{\mu},\boldsymbol{\Sigma})$ by maximizing the ELBO with randomly initialized $\boldsymbol{\Sigma}$.

## 2.2 BAYESIAN FINE-TUNING IN THE STYLE OF FINE-TUNING DNNS

We develop practical learning algorithms under the stochastic VI scheme to fine-tune the imperfect variational posterior, and to cope with contemporary ML frameworks. In the following, we discuss how to deal with each term in Eq. (2). Algorithm 1 gives an overview of *BayesAdapter*.

**Complexity loss $\mathcal{L}_c$.** Without losing generality, we assume an isotropic Gaussian prior $p(\boldsymbol{w}) = \mathcal{N}(\boldsymbol{w};\boldsymbol{0},\sigma_0^2\mathbf{I})$. Then the complexity loss is derived as:

$$\mathcal{L}_c = -\frac{1}{n}D_{\mathrm{KL}}\left(\mathcal{N}(\boldsymbol{w};\boldsymbol{\mu},\boldsymbol{\Sigma})\|\mathcal{N}(\boldsymbol{w};\boldsymbol{0},\sigma_0^2\mathbf{I})\right) = -\frac{\boldsymbol{\mu}^T\boldsymbol{\mu} + \mathrm{tr}(\boldsymbol{\Sigma})}{2\sigma_0^2 n} + \frac{\log\det\boldsymbol{\Sigma}}{2n} + c, \tag{4}$$

where $\mathrm{tr}$ and $\det$ are matrix trace and determinant, respectively. $c$ is a constant. The gradients of $\mathcal{L}_c$ w.r.t. $\boldsymbol{\mu}$ and $\boldsymbol{\Sigma}$ can be estimated precisely as:

$$\nabla_{\boldsymbol{\mu}}\mathcal{L}_c = -\frac{\boldsymbol{\mu}}{\sigma_0^2 n}, \quad \nabla_{\boldsymbol{\Sigma}}\mathcal{L}_c = \frac{\sigma_0^2\boldsymbol{\Sigma}^{-1} - \mathbf{I}}{2\sigma_0^2 n}. \tag{5}$$

Eq. (5) indicates that $\max_{\boldsymbol{\mu}}\mathcal{L}_c$ amounts to applying a weight decay regularizer with coefficient $\lambda = \frac{1}{\sigma_0^2 n}$ on $\boldsymbol{\mu}$, which can be conveniently optimized by leveraging the built-in weight decay modules in ML frameworks such as TensorFlow (Abadi et al., 2016) or PyTorch (Paszke et al., 2019).

Directly computing $\nabla_{\boldsymbol{\Sigma}}\mathcal{L}_c$ involves matrix inversion. Implementing the posterior as matrix-variate Gaussian is an alternative, while existing algorithms for matrix-variate Gaussian posterior typically exhibit high complexity in time or memory, limited compatibility with contemporary NN building block operations (e.g., convolution), and struggle to scale with data-parallel distributed training (Louizos & Welling, 2016; Sun et al., 2017; Osawa et al., 2019b). To simplify the implementation and boost scalability, we assume a fully factorized Gaussian variational by devising $\boldsymbol{\Sigma}$ as

---
[2]We use BNNs equivalently with variational BNNs in the following text when there is no ambiguity.

$\text{diag}(\exp(2\boldsymbol{\psi}))$, where $\boldsymbol{\psi} \in \mathbb{R}^p$ is the parameter to be optimized along with $\boldsymbol{\mu}$ (i.e., $\boldsymbol{\theta} = (\boldsymbol{\mu}, \boldsymbol{\psi})$). Injecting this into Eq. (5) gets a more concise gradient estimator: $\nabla_{\boldsymbol{\psi}} \mathcal{L}_c = \mathbf{1}/n - \lambda \exp(2\boldsymbol{\psi})$, meaning that $\max_{\boldsymbol{\psi}} \mathcal{L}_c$ adds an exponential weight decay of $\boldsymbol{\psi}$ with coefficient $\lambda$, which can be realized by modifying only two lines of code on top of de facto DL frameworks (see Figure 1).

**Expected log-likelihood** $\mathcal{L}_{\text{ell}}$. With the complexity loss expressed as weight decay, we now develop efficient ways for calculating the $\mathcal{L}_{\text{ell}}$ at the end of forward pass, and for performing backpropagation afterwards. In particular, we derive a Monte Carlo (MC) estimation of $\mathcal{L}_{ell}$ based on reparameterization (Kingma & Welling, 2013): we sample a $p$-dimensional Gaussian noise $\boldsymbol{\epsilon} \sim \mathcal{N}(\mathbf{0}, \mathbf{I})$, then obtain the sampled parameter for the whole mini-batch $\mathcal{B}$ of data via $\boldsymbol{w} = \boldsymbol{\mu} + \exp(\boldsymbol{\psi})\boldsymbol{\epsilon}$, given which we approximate $\mathcal{L}_{ell}$ with $\mathcal{L}'_{ell} = \frac{1}{|\mathcal{B}|} \sum_{(\boldsymbol{x}_i, y_i) \in \mathcal{B}} \log p(y_i | \boldsymbol{x}_i; \boldsymbol{w})$. The gradients of $\boldsymbol{\mu}$ and $\boldsymbol{\psi}$ can be derived automatically with autodiff libraries, thus the training resembles that of normal DNNs.

However, gradients derived by $\mathcal{L}'_{ell}$ might exhibit high variance, caused by sharing one set of sampled parameters $\boldsymbol{w}$ across all the training instances in $\mathcal{B}$. *Local reparameterization* is proposed to reduce the variance, but it requires at least 2x forward-backward FLOPS than vanilla reparameterization (refer to Kingma et al. (2015) for more details). Flipout (Wen et al., 2018) is an alternative solution. But it is only suitable for perturbation based MC estimation and its modeling assumptions make Flipout unable to handle complex variational posterior like a FLOW (Louizos & Welling, 2017), or an implicit model (Shi et al., 2018b). Besides, it is still as slow as *local reparameterization*. To mitigate these issues, we propose *exemplar reparametrization* (ER) which samples a separate set of parameters for every exemplar in the minibatch. Formally, for $\forall \boldsymbol{x}_i \in \mathcal{B}$, we draw $\boldsymbol{w}^{(i)} = \boldsymbol{\mu} + \exp(\boldsymbol{\psi})\boldsymbol{\epsilon}^{(i)}$ where $\boldsymbol{\epsilon}^{(i)} \sim \mathcal{N}(\mathbf{0}, \mathbf{I})$, and approximate the *expected log-likelihood* by $\mathcal{L}^*_{ell} = \frac{1}{|\mathcal{B}|} \sum_{(\boldsymbol{x}_i, y_i) \in \mathcal{B}} \log p(y_i | \boldsymbol{x}_i; \boldsymbol{w}^{(i)})$.

Obviously, ER is distribution agnostic, and is readily applicable to various variational distributions. While ER generates more parameters at training, they are mostly temporary, and the resultant computational FLOPS are provably identical to that of the vanilla reparameterization. The challenge of ER is to cope with nowadays ML frameworks and maintain computing efficiency, because off-the-shelf computation kernels in autodiff libraries typically assume a batch of instances share a common set of parameters. We present an example in Figure 2 on how the standard `convolution` op can be converted into its exemplar version without compromising computational efficiency. The key insight here is that multiple exemplar convolutions can be expressed as a group convolution, which can be performed in parallel using a single group convolution kernel, leveraging the optimized implementations provided by various device-propriety kernel backends (e.g. cuDNN (Chetlur et al., 2014)). Other common operators such as matrix multiplication are straightforward to handle (refer to Appendix A).

```
# assume shape x: [b, i, h, w]; w, mu, psi: [o, i, k, k]
def DNN_conv(x, w, stride, padding, groups):
    return conv2d(x, w, stride, padding, groups)

def BayesAdapter_conv(x, mu, psi, stride, padding, groups):
    b = x.shape[0]
    # sample a batch of parameters w: [b, o, i, k, k]
    w = mu + exp(psi) * randn(b, *list(mu.shape))
    # reshape w to have shape [b*o, i, k, k]
    w = w.flatten(start_dim=0, end_dim=1)
    # reshape x to have shape [1, b*i, h, w]
    x = x.flatten(start_dim=0, end_dim=1).unsqueeze(0)
    # perform b convs in parallel
    y = conv2d(x, w, stride, padding, groups*b)
    # reshape the result to standard format
    return y.view(b, mu.shape[0], y.shape[2], y.shape[3])
```

**Figure 2:** The comparison between standard convolution and convolution used in *BayesAdapter*.

With this insight, *BayesAdapter* enables to obtain a BNN with only minor computational cost in addition to pre-training, and can immediately benefit from the availability of higher-performance computational kernels (e.g., more powerful group convolution kernel).

## 3 CALIBRATE THE UNCERTAINTY ESTIMATION

So far we have developed an inexpensive fine-tuning procedure to obtain BNNs from deterministic NNs. While BNNs can offer uncertainty estimates, these uncertainty measures are highly non-smooth due to the non-convexity of NNs – they might exhibit high uncertainty for data from faraway out-of-distribution (OOD) regions, but become vulnerable on OOD samples close to the normal ones (Grosse et al., 2018), rendering BNNs unable to react to potentially harmful inputs. We quantify this phenomenon in Section 4.2. To address this problem, we next develop methods to further calibrate the uncertainty estimation of naively trained BNNs. Inspired by recent work on OOD detection (Wang et al., 2020; Durall et al., 2020), we propose to additionally incorporate uncertainty regularization on top of the above fine-tuning procedure. The idea is to force BNNs to generate inconsistent predictions for each sample from a cheaply collected OOD sample set, so that they acquire the ability to yield high uncertainty for OOD samples with similar fingerprints.

---

**Algorithm 1:** BayesAdapter

---

**Input:** normal training set $\mathcal{D}$ in the size of $n$, OOD training set $\mathcal{D}^{\dagger}$, weight decay coefficient $\lambda$ for both the pre-training and the fine-tuning, threshold $\gamma$, learning rates $lr_{\boldsymbol{\mu}}$, $lr_{\boldsymbol{\psi}}$, fine-tuning epochs $T$

1 Pre-train the DNN counterpart of the target BNN on $\mathcal{D}$ by MAP; denote the converged parameters as $\boldsymbol{\mu}$
2 Create randomly initialized parameters $\boldsymbol{\psi}$; make the computation modules be Bayesian (see Figure 2)
3 Build optimizers $\text{opt}_{\boldsymbol{\mu}}$ and $\text{opt}_{\boldsymbol{\psi}}$ (see Figure 1) with learning rate $lr_{\boldsymbol{\mu}}$ and $lr_{\boldsymbol{\psi}}$ for $\boldsymbol{\mu}$ and $\boldsymbol{\psi}$ respectively
4 **for** *epoch = 1, 2, ..., T* **do**
5     **for** *mini-batch* $\mathcal{B} = \{(\boldsymbol{x}_i, y_i)\}_{i=1}^{|\mathcal{B}|}$ *in* $\mathcal{D}$, *mini-batch* $\mathcal{B}^{\dagger} = \{\boldsymbol{x}_i^{\dagger}\}_{i=1}^{|\mathcal{B}^{\dagger}|}$ *in* $\mathcal{D}^{\dagger}$ **do**
6        Build the whole mini-batch $\{\boldsymbol{x}_1, ..., \boldsymbol{x}_{\mathcal{B}}, \boldsymbol{x}_1^{\dagger}, ..., \boldsymbol{x}_{|\mathcal{B}^{\dagger}|}^{\dagger}, \boldsymbol{x}_1^{\dagger}, ..., \boldsymbol{x}_{|\mathcal{B}^{\dagger}|}^{\dagger}\}$, and feed it into the model
7        Given the predictive distribution and labels $\{y_i\}_{i=1}^{|\mathcal{B}|}$, compute $\mathcal{L}_{ell}^{*}$ and $\mathcal{L}_{\text{unc}}$
8        Derive the gradients of $\mathcal{L}_{ell}^{*} + \mathcal{L}_{\text{unc}}$ w.r.t. $\boldsymbol{\mu}$ and $\boldsymbol{\psi}$ via AutoGrad
9        Update the parameters $\boldsymbol{\mu}$ and $\boldsymbol{\psi}$ with optimizers $\text{opt}_{\boldsymbol{\mu}}$ and $\text{opt}_{\boldsymbol{\psi}}$

---

To achieve this, we start by defining a differentiable uncertainty metric in terms of mutual information, following Smith & Gal (2018):

$$\mathcal{I}(\boldsymbol{w}, y|\boldsymbol{x}, \mathcal{D}) \approx H\left(\frac{1}{S}\sum_{s=1}^{S} p(y|\boldsymbol{x}; \boldsymbol{w}^{(s)})\right) - \frac{1}{S}\sum_{s=1}^{S} H\left(p(y|\boldsymbol{x}; \boldsymbol{w}^{(s)})\right), \text{where } \boldsymbol{w}^{(s)} \sim \mathcal{N}(\boldsymbol{w}; \boldsymbol{\mu}, \boldsymbol{\psi}), s = 1, ..., S. \quad (6)$$

$H$ is the Shannon entropy. $\mathcal{I}$ highly correlates with softmax variance (Smith & Gal, 2018), and measures the *epistemic uncertainty* which describes uncertainty in the model and can be used to identify OOD instances. Then, assuming access to an OOD dataset $\mathcal{D}^{\dagger} = \{\boldsymbol{x}_i^{\dagger}\}_{i=1}^{n^{\dagger}}$, we enforce the model to behave uncertainly on each of them by optimizing a margin loss with threshold $\gamma$:

$$\max_{\theta} \mathcal{L}_{\text{unc}} = \frac{1}{|\mathcal{B}^{\dagger}|} \sum_{\boldsymbol{x}_i^{\dagger} \in \mathcal{B}^{\dagger}} \min\left(\mathcal{I}(\boldsymbol{w}, y|\boldsymbol{x}_i^{\dagger}, \mathcal{D}), \gamma\right), \quad (7)$$

where $\mathcal{B}^{\dagger}$ refers to a mini-batch of OOD data. For efficiency, we adopt $S = 2$ MC samples for estimating $\mathcal{I}(\boldsymbol{w}, y|\boldsymbol{x}_i^{\dagger}, \mathcal{D})$ in Eq. (7) in the training. While this loss has a seemingly opposite form from the consistency-promoting loss in semi-supervised learning (SSL) (Laine & Aila, 2016), they share the same design philosophy: $\mathcal{L}_{\text{unc}}$ maximizes the prediction inconsistency of OOD instances so as to distinguish them from in-distribution instances, while SSL minimizes the prediction inconsistency of unlabeled data so to classify them without labels. Put it in the context of autonomous driving: if the model is trained on data only containing scenes in regular weather, we can take a small set of scene data of extreme weather, e.g., tornado and sandstorm, to regularize the training following Eq. (7). Then the model will learn to identify these abnormal scenes based on predictive uncertainty, thus can refuse to make unreliable decisions in these scenes.

Constructing the OOD dataset $\mathcal{D}^{\dagger}$ is flexible and application-specific. In discriminative tasks, two types of OOD data of particular concerns are adversarial and fake samples, which can be both collected trivially following procedures described below.

**Adversarial samples.** Directly generating adversary samples following methods like PGD (Madry et al., 2017) might be expensive. We propose a more cost-effective alternative based on a key observation: given a valid perturbation space $[-\delta_m, \delta_m]^d$ where $\delta_m$ is the maximum norm under the $l_{\infty}$ threat model, we can see that uniform noises $\boldsymbol{\delta} \sim \mathcal{U}(-\delta_m, \delta_m)^d$ radically encompass the adversarial perturbations which usually reside at local optimas. Thus we can add uniformly perturbed samples into uncertainty training to direct the model to behave uncertainly on randomly contaminated data, bypassing the potential cost of generating real adversary samples. The results in Sec 4.2 surprisingly confirm the effectiveness of uniform noises, and imply a strong connection between uniform noises and adversarial ones, which deserves a future investigation.

**Fake samples.** Fake samples can be obtained by utilizing pretrained state-of-the-art GANs (Miyato et al., 2018; Brock et al., 2018), DeepFake (Deepfakes, 2018), and FaceSwap (Faceswap, 2018). We use only 1000 random fake samples for *Bayesian fine-tuning* on diverse benchmarks.

For both, we empirically find the proposed uncertainty regularization is *data efficient* – with access to a proxy set of adversarial samples and a small set of fake samples, the model can acquire reliable, transferable uncertainty quantification.

## 4 EXPERIMENTS

In this section, we evaluate *BayesAdapter* on a diverse set of challenging benchmarks.

**Table 1:** Predictive performance comparison on image classification. NLL denotes the negative log-likelihood.

| Method | CIFAR-10 (wide-ResNet-28-10) | | ImageNet (ResNet-50) | | |
|---|---|---|---|---|---|
| | TOP1 (%) ↑ | NLL ↓ | TOP1 (%) ↑ | TOP5 (%) ↑ | NLL ↓ |
| *MAP* | 96.92 | 0.1312 | 76.13 | 92.86 | 0.9618 |
| *Deep Ensemble* | **97.40** | **0.0869** | - | - | - |
| *SWAG* | 96.32 | 0.1122 | - | - | - |
| *Laplace Approx.* | 96.41 | 0.1204 | 75.89 | 92.70 | 0.9739 |
| *MC dropout* | 96.95 | 0.1151 | 74.88 | 92.32 | 0.9884 |
| *BNN* | 97.02 | 0.0975 | 75.97 | 92.95 | 0.9435 |
| *BayesAdapter-* | **97.09** | **0.0945** | **76.49** | **93.10** | **0.9337** |
| *BayesAdapter* | 96.82±0.07 | 0.1004±0.0026 | 76.26±0.06 | 92.96±0.03 | 0.9428±0.0020 |

**Table 2:** Accuracy ↑ comparison on open-set face recognition with MobileNetV2 architecture.

| Method | LFW | CPLFW | CALFW | CFP-FF | CFP-FP | VGGFace2 | AgeDB-30 |
|---|---|---|---|---|---|---|---|
| *MAP* | 98.2% | 84.0% | **87.6%** | 97.8% | 92.7% | 91.7% | 85.3% |
| *MC dropout* | 98.2% | 83.6% | 87.3% | 97.8% | 92.8% | **92.6%** | **86.0%** |
| *BNN* | 97.8% | 82.4% | 85.7% | 96.8% | 91.4% | 90.5% | 83.8% |
| *BayesAdapter-* | 98.4% | **84.4%** | 87.3% | 97.7% | 92.7% | 92.3% | 85.1% |
| *BayesAdapter* | **98.5%** | 84.1% | 87.4% | **97.9%** | **93.1%** | 92.5% | 84.8% |

**Settings.** We first conduct experiments on CIFAR-10 (Krizhevsky et al., 2009) and ImageNet (Deng et al., 2009) using wide-ResNet-28-10 (Zagoruyko & Komodakis, 2016) and ResNet-50 (He et al., 2016), respectively. Besides, we train face recognition models on CASIA (Yi et al., 2014) with MobileNetV2 (Sandler et al., 2018), and perform comprehensive evaluation on face verification datasets including LFW (Huang et al., 2007), CPLFW (Zheng & Deng, 2018), CALFW (Zheng et al., 2017), CFP (Sengupta et al., 2016), VGGFace2 (Cao et al., 2018), and AgeDB-30 (Moschoglou et al., 2017). We pre-train DNNs following standard protocols, and perform *Bayesian fine-tuning* for 40, 12, and 16 epochs with weight decay coefficients (i.e., $\lambda$) 2e-4, 1e-4, and 5e-4 on these benchmarks respectively. We set uncertainty threshold $\gamma$ according to an observation that the normal examples usually present $< 0.75$ mutual information uncertainty, across the studied scenarios. Then we use $\gamma = 0.75$ in the regularization to push both the adversarial and fake data to exhibit distinguishable uncertainty from the normal data. We initialize $\psi$ uniformly from $[-6, -5]^p$ and use 20-step PGD as validation adversaries. On the three benchmarks, the perturbation budgets $\delta_m$ are set 0.031, 16/255, and 16/255, and the fake samples are obtained from SNGAN (Miyato et al., 2018), Big-GAN (Brock et al., 2018), and DeepFake. We perform intensive data augmentation for fake training data with a random strategy including Gaussian blur, JPEG compression, *etc.* We defer more details to Appendix B. We run every experiment 3 times on 8 RTX 2080-TIs and report the average.

**Baselines.** We compare the full *BayesAdapter* to extensive baselines including: (1) *MAP*, (2) *Laplace Approx.*: Laplace Approximation with diagonal Fisher information matrix, (3) *MC dropout* (detailed in Appendix B), (4) *BNN*: BNNs trained from scratch by solving Eq. (2) without uncertainty regularization, (5) *BayesAdapter-*: a variant of *BayesAdapter* without uncertainty regularization. We also include *Deep Ensemble* (Lakshminarayanan et al., 2017), one of the state-of-the-art BNNs, and *SWAG* (Maddox et al., 2019), whose performance is not worse than SGLD (Welling & Teh, 2011), KFAC Laplace (Ritter et al., 2018), and temperature scaling (Guo et al., 2017), into the comparison on CIFAR-10[3].

**Metrics.** We concern (*i*) the *posterior predictive* performance with $S = 100$ MC samples; (*ii*) the average precision (AP) of directly using the uncertainty estimated by Eq. (6) ($S = 20$) to distinguish OOD test samples (labeled 1) from normal test samples (labeled 0). Eq. (6) of the deterministic baseline *MAP* is 0, so we take the predictive entropy as an alternative uncertainty measure for *MAP*.

### 4.1 Predictive Performance

We compare the prediction performance, which is of central importance in practice, of various methods in Table 1 and 2. *Deep Ensemble* shows outperforming classification performance because the ensemble candidates can investigate diverse function modes, but it is orders of magnitude more expensive than *BayesAdapter*. *BayesAdapter-* notably surpasses the *MAP*, especially in NLL, verifying the modeling superiority of a Bayesian formulation and highlights the practical value of our workflow. *Laplace Approx.* is consistently worse than *MAP*. In all settings, *BNN* is significantly defeated by *BayesAdapter-*, confirming our claim that performing *Bayesian fine-tuning* from the

---

[3]Currently, we have not scaled *Deep Ensemble* and *SWAG* up to ImageNet due to resource constraints.

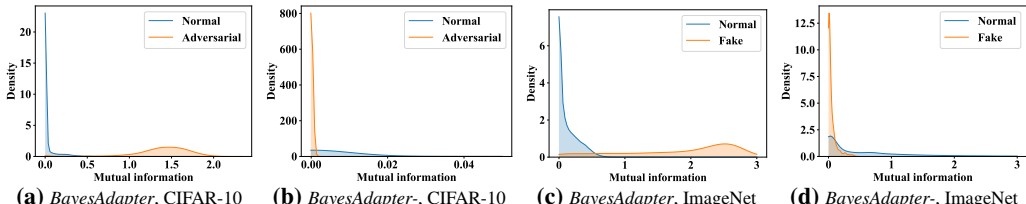

**Figure 3:** The histograms for the mutual information uncertainty of normal data and OOD data given by models trained w/ and w/o uncertainty regularization.

**Table 3:** Comparison on the quality of uncertainty estimates in terms of average precision (AP) ↑ of uncertainty based binary classification. (CIFAR-10 and ImageNet)

| Method | CIFAR-10 | | ImageNet | |
|---|---|---|---|---|
| | Adversarial (PGD) | Fake (SNGAN) | Adversarial (PGD) | Fake (BigGAN) |
| *MAP* | 0.307 | 0.800 | 0.308 | 0.010 |
| *Deep Ensemble* | 0.427 | 0.812 | - | - |
| *SWAG* | 0.316 | 0.816 | - | - |
| *Laplace Approx.* | 0.308 | 0.800 | 0.311 | 0.015 |
| *MC dropout* | 0.308 | 0.803 | 0.309 | 0.012 |
| *BNN* | 0.307 | 0.799 | 0.310 | 0.021 |
| *BayesAdapter-* | 0.307 | 0.806 | 0.387 | 0.013 |
| *BayesAdapter* | **0.993**±0.003 | **0.994**±0.001 | **0.964**±0.009 | **0.848**±0.037 |

converged deterministic checkpoints is beneficial to bypass the local optimas potentially encountered by direct Bayesian inference. The popular baselines *MC dropout* and *SWAG* show weaker performance on ImageNet and CIFAR-10, respectively, revealing limited applicability. Also of note that no method shows dominant performance on face recognition, probably due to the diversity of these validation sets. Across these tasks, *BayesAdapter* is slightly worse than its regularization-free version *BayesAdapter-*. This is reasonable since such a regularization enforces the model to trade partial capacity for fidelity of uncertainty estimates. Nevertheless, *BayesAdapter* is substantially better than its fine-tuning start point *MAP* and the *BNN* trained from scratch in most settings.

**Speedup.** BayesAdapter is a much more economical way to obtain BNNs. To interpret the speedup of *BayesAdapter* over *BNN*, we assume deterministic ResNet-50 takes one unit time $t$ for one epoch of training on ImageNet, and observe Bayesian ResNet-50 takes $\approx 2.1t$ for one epoch. Thus, *BNN* trained from scratch consumes $189t$ for 90-epoch training, while *BayesAdapter-* need $t*90+2.1t*12=115.2t$, saving $73.8t$ (around **40%**) training time than *BNN*. [4]

### 4.2 QUALITY OF UNCERTAINTY ESTIMATES

We study the effects of the proposed uncertainty regularization by visualizing the predictive uncertainty of *BayesAdapter* and *BayesAdapter-* on validation data in Figure 3. On both CIFAR-10 and ImageNet, *BayesAdapter* yields evidently higher uncertainty for OOD data than normal data, while *BayesAdapter-* is on the contrary, showing it can effectively calibrate the predictive uncertainty.

To precisely evaluate its efficacy, we quantitatively assess the quality of the predictive uncertainty of various methods by estimating AP, which reflects if *the model knows what it knows*. As stated, we take adversarial samples crafted by PGD and fake samples from GANs and DeepFake as proxies of harmful OOD data. We report the results in Table 3 and Table 6 in Appendix C. As shown, *SWAG*, *Laplace Approx.*, *MC dropout*, *BNN*, and *BayesAdapter-* perform all as bad as *MAP* across settings, except that *MC dropout* is capable of partially detecting OOD data on face tasks[5]. Despite impressive prediction accuracy, *Deep Ensemble* also yields unreliable uncertainty estimates on these two kinds of challenging OOD data. These results echo our concern on the reliability of existing BNNs' predictive uncertainty. By contrast, *BayesAdapter*, which is fine-tuned upon *MAP* for only several rounds based on low-cost supervisions, achieves near-perfect results in detecting OOD instances on CIFAR-10 and face recognition, and also detects most of the OOD instances on ImageNet (see Appendix D for some samples).

### 4.3 ABLATION STUDY

---

[4]In practice, *BayesAdapter* would be a little slower than *BayesAdapter-* due to the incorporation of the OOD training set, but still much more efficient than *BNN*.

[5]We speculate this may relate to where to add dropout in the NN architecture, but leave it for future study.

**Model calibration.** Model calibration is another important aspect of the uncertainty estimation. Suggested by pioneering works, we take the *Expected Calibration Error* (ECE) (Guo et al., 2017) as the measure of calibration, and report the ECE of the studied methods in Table 4. The ECE of *BayesAdapter* is on par with the *MC dropout*, *Deep Ensemble*, and *BNN* baselines, but *BayesAdapter* can meanwhile offer much better uncertainty for detecting risky OOD data.

**Table 4:** Comparison on ECE ↓.

| Method | CIFAR-10 | ImageNet |
|---|---|---|
| *MAP* | 0.0198 | 0.0373 |
| *Deep Ensemble* | **0.0057** | - |
| *SWAG* | 0.0088 | - |
| *Laplace Approx.* | 0.0106 | 0.0375 |
| *MC dropout* | 0.0119 | **0.0152** |
| *BNN* | **0.0055** | 0.0183 |
| *BayesAdapter-* | 0.0094 | 0.0159 |
| *BayesAdapter* | **0.0057** | 0.0165 |

**Transferability of uncertainty quantification.** One may wonder if the uncertainty quantification learned according to specialized OOD data can generalize to other OOD data. To figure out this problem, we evaluate the *BayesAdapter* trained on CIFAR-10 on 10000 samples from PGGAN (Karras et al., 2017) whose patterns are unseen during training. We compute their uncertainty and calculate the AP metric, obtaining **0.932**. As comparison, the APs of *MAP*, *Deep Ensemble*, *SWAG*, *Laplace Approx.*, *MC dropout*, *BNN*, *BayesAdapter-* on such data are 0.789, 0.797, 0.809, 0.800, 0.792, 0.793, 0.803 respectively. On the other hand, we craft adversarial examples by the fast gradient sign method (FGSM) (Goodfellow et al., 2014) against the ResNet-152 DNN model with 1000 validation images from ImageNet. Then we estimate the AP on these instances, and obtain 0.011, 0.125, 0.027, 0.202, 0.019, and **0.882** for *MAP*, *Laplace Approx.*, *MC dropout*, *BNN*, *BayesAdapter-*, and *BayesAdapter* respectively. These studies validate the transferability of our uncertainty estimation.

**The effectiveness of exemplar reparameterization.** We build a toy model with only a Bayesian convolutional layer, fixing the model input and target output, and computing the variance of stochastic gradients across 500 runs. We average the gradient variance of $\mu$ and $\psi$ over all their coordinates, and observe that standard reparameterization typically introduces more than $100\times$ variance than *exemplar reparameterization*, despite with the same FLOPS.

**Ablation study on uncertainty threshold $\gamma$.** We perform an ablation study regarding $\gamma$ on CIFAR-10 to evaluate the hyper-parameter tolerance of the proposed method. Table 5 presents the results. The results reveal that values of $\gamma \in [0.75, 1.0]$ may be good choices for OOD detection, and also echo the observation that normal examples usually present $< 0.75$ uncertainty.

**Table 5:** Ablation study on $\gamma$.

| $\gamma$ | Acc. | AP (PGD) | AP (fake) |
|---|---|---|---|
| 0.25 | 96.93% | 0.915 | 0.910 |
| 0.50 | 96.70% | 0.948 | 0.981 |
| 0.75 | 96.82% | 0.993 | 0.994 |
| 1.00 | 96.74% | 0.991 | 0.994 |
| 1.50 | 96.79% | 0.944 | 0.988 |

**The impacts of ensemble number.** We draw the change of test accuracy w.r.t. the number of MC samples $S$ for estimating Eq. (3) in Figure 4. The model is trained by *BayesAdapter* on ImageNet. The points on the red line represent the individual accuracies of the 100 parameter samples. The yellow dashed line refers to the deterministic inference with only the Gaussian mean. The green line displays the effects of *Bayes ensemble* – the predictive performance increases from $< 74\%$ to $> 76\%$ quickly before seeing 20 parameter samples, and gradually saturate after that. That is why we use 20 samples for estimating uncertainty and crafting adversarial samples.

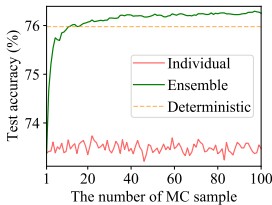

**Figure 4:** The change of test accuracy w.r.t. the number of MC samples for estimating Eq. (3).

**Uncertainty-based rejective decision.** In practice, we expect our models can reject to predict for data with relatively large uncertainty, and only care about the data that they are certain about. In this spirit, we sort the validation data of ImageNet w.r.t. the uncertainty provided by *BayesAdapter*, and split them into 10 buckets of equal size. We depict the average accuracy of each bucket in Figure 5. As expected, our BNN is more accurate for instances with smaller uncertainty. Quantitatively, there are 95% instances with uncertainty less than 0.45, and their accuracy is 78.6%; there are 90% instances with uncertainty less than 0.37, and their accuracy is 80.7%; there are 80% instances with uncertainty less than 0.25, and their accuracy is 84.8%.

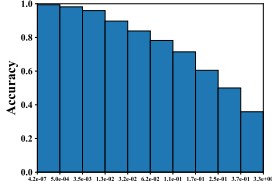

**Figure 5:** Comparison on accuracy for instance buckets of equal size but with rising uncertainty.

## 5 RELATED WORK

Fruitful works have emerged in the BNN community in the last decade (Graves, 2011; Welling & Teh, 2011; Blundell et al., 2015; Kingma & Welling, 2013; Balan et al., 2015; Liu & Wang, 2016; Kendall & Gal, 2017). However, most of the existing works cannot achieve the goal of practicability. For example, Liu & Wang (2016); Louizos & Welling (2016; 2017); Shi et al. (2018a); Sun et al. (2019) trade learning efficiency for flexible variational posteriors, leading to restrictive scalability. Khan et al.; Zhang et al.; Osawa et al. build Adam-like optimizers to do variational inference, but their parallel training throughput and compatibility with data augmentation are inferior to SGD. Empirical Bayes methods, e.g., Monte Carlo (MC) dropout (Gal & Ghahramani, 2016), deep ensemble (Lakshminarayanan et al., 2017), and SWAG (Maddox et al., 2019), usually maintain impressive predictive performance, but suffer from degenerated uncertainty estimates (Fort et al., 2019) or expensive training/storage cost. What's worse, the existing works usually evaluate on impractical OOD data (Louizos & Welling, 2017; Pawlowski et al., 2017) to show the promise of Bayesian principle. Instead, we offer a new evaluation standard in this work, which may benefit the following works.

Laplacian approximation (Bleistein & Handelsman, 1986; Ritter et al., 2018) is a known approach to transform a DNN to a BNN, but it is inflexible due to its postprocessing nature and some strong assumptions made for practical concerns. Alternatively, *BayesAdapter* works in the style of fine-tuning, which is more natural and economical for deep networks. Bayesian modeling the last layer of a DNN is proposed recently (Kristiadi et al., 2020), and its combination with *BayesAdapter* deserves an investigation. *BayesAdapter* connects to MOPED (Krishnan et al.) in that their variational configurations are both based on *MAP*. Yet, beyond MOPED, *BayesAdapter* is further designed to achieve good user-friendliness, improved learning stability, and trustable uncertainty estimation, by virtue of optimizers with built-in weight decay, exemplar reparameterization, and uncertainty regularization, respectively, which significantly boost the practicability of *BayesAdapter*, especially in real-world and large-scale settings.

## 6 CONCLUSION

In this work, we propose a scalable *BayesAdapter* framework to learn practical BNNs. Our core idea is to learn a BNN by first pre-training its DNN counterpart and then performing *Bayesian fine-tuning*. In BayesAdapter, we develop a plug-and-play instantiation of stochastic VI, and propose *exemplar reparameterization* to reduce the gradient variance. We also propose a generic uncertainty regularization to calibrate the uncertainty quantification given low-cost supervisions. We evaluate *BayesAdapter* in diverse realistic scenarios and report promising results.

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

## A  THE EXEMPLAR VERSION OF POPULAR OPERATORS

As introduced in Sec 2.2, the regular convolution can be elegantly converted into an exemplar version by resorting to group convolution. The other popular operators are relatively easy to handle. For example, we substitute the qualified batch matrix multiplication, which is highly optimized in the well-known autodiff libraries, for matrix multiplication. For the affine transformation in batch normalization (Ioffe & Szegedy, 2015), we can at first sample dedicated affine weight and bias for every exemplar in the batch, then perform transformation with these two batches of parameters by just not *broadcasting* on the batch dimension.

## B  MORE EXPERIMENTAL DETAILS

The only two important hyper-parameters are the weight decay coefficient $\lambda$ and the uncertainty threshold $\gamma$. Other hyper-parameters for defining PGD or specifying learning rates, etc., all follow standard practice in the DL community. The number of fake data training (1000) and the number of MC samples for evaluation ($S$) are flexible and not tuned.

For $\lambda$, we keep it consistent between pre-training and fine-tuning (stated in Algorithm 1), without elaborated tuning, for example, $\lambda = 2e - 4$ for the wide-ResNet-28-10 architecture on CIFAR-10, $\lambda = 1e - 4$ for ResNet-50 architecture on ImageNet, and $\lambda = 5e - 4$ for MobileNet-V2 architecture on CASIA. These values correspond to isotropic Gaussian priors with $\sigma_0^2$ as 0.1, 0.0078, and 0.0041 on CIFAR-10, ImageNet, and CASIA, respectively. It is notable that for a "small" dataset like CIFAR-10, a flatter prior is preferred. While on larger datasets with stronger data evidence, we need a sharper prior for regularization.

For $\gamma$, we use $\gamma = 0.75$ for training across all the scenarios. But it is not used for OOD detection in the testing phase. For estimating the results of OOD detection, we use the non-parametric metric average precision (see the metric part of Section 4), which is the Area Under the Precision-Recall Curve and is more suitable than the ROC-AUC metric when there is class imbalance.

For the pre-training, we follow standard protocols available online. On CIFAR-10, we perform CutOut (DeVries & Taylor, 2017) transformation upon popular resize/crop/flip transformation for data augmentation. On ImageNet, we leverage the ResNet-50 checkpoint on PyTorch Hub as the converged deterministic model. On face tasks, we train MobileNetV2 following popular hyper-parameter settings, and the pre-training takes 90 epochs. We use the same weight decay coefficients in both the pre-training and the fine-tuning.

For the fine-tuning, we set $lr_\psi$ to decay at 1/4, 1/2, and 3/4 of the total fine-tuning steps from 0.1, and set $lr_\mu$ to be the final value of $lr_\psi$ on the CIFAR-10, ImageNet, and face recognition benchmarks. We add a coefficient 3 before the $\mathcal{L}_{\mathrm{unc}}$ term in Line 8 of Algorithm 1 for *Bayesian fine-tuning* on ImageNet to achieve better uncertainty calibration. For models on face recognition, we utilize the features before the last FC layer of the MobileNetV2 architecture to conduct feature distance-based face classification in the validation phase, due to the open-set nature of the validation data. The *Bayes ensemble* is similarly achieved by assembling features from multiple runs as the final feature for estimating predictive performance. But we still adopt the output from the last FC layer for uncertainty estimation (i.e., calculating Eq. (6)).

The training perturbation budget is identical to the evaluation budget on CIFAR-10 and ImageNet. But we set the budget of the uniform noise used for training in face tasks to be 1/4 of the evaluation budget to make the models more sensitive to the perturbed data. We adopt PGD for generating adversarial samples in the validation phase. Concretely, we attack the *posterior predictive* objective, i.e., Eq. (3), with $S = 20$ MC samples. On CIFAR-10, we set $\delta_m = 0.031$ and perform PGD for 20 steps with step size at 0.003. On ImageNet and face recognition, we set $\delta_m = 16/255$ and perform PGD for 20 steps with step size at $1/255$.

Regarding the fake data, we craft 1000 fake samples for training and 10000 ones for evaluation with SNGAN (Miyato et al., 2018) on CIFAR-10; we craft 1000 fake samples for training and 1000 ones for evaluation with BigGAN (Brock et al., 2018) on ImageNet; we randomly sample 1000 fake samples for training and 10000 ones for evaluation from DeepFakes (Deepfakes, 2018), FaceSwap (Faceswap, 2018) and Face2Face (Thies et al., 2016) on face recognition.

**Table 6:** Comparison on the quality of uncertainty estimates in terms of AP ↑ on face recognition tasks. The upper part is for adversarial instances and the other part is for DeepFake.

| Method | LFW | CPLFW | CALFW | CFP_FF | CFP_FP | VGG2_FP | AGEDB_30 |
|---|---|---|---|---|---|---|---|
| MAP | 0.191 | 0.192 | 0.191 | 0.211 | 0.205 | 0.200 | 0.199 |
| MC dropout | 0.965 | 0.946 | 0.959 | 0.965 | 0.949 | 0.954 | 0.952 |
| BNN | 0.399 | 0.282 | 0.429 | 0.390 | 0.271 | 0.291 | 0.327 |
| BayesAdapter- | 0.189 | 0.189 | 0.189 | 0.193 | 0.191 | 0.191 | 0.190 |
| BayesAdapter | **0.998** | **0.981** | **0.999** | **0.999** | **0.983** | **0.990** | **0.995** |
| MAP | 0.389 | 0.456 | 0.375 | 0.394 | 0.454 | 0.519 | 0.437 |
| MC dropout | 0.846 | 0.664 | 0.862 | 0.874 | 0.685 | 0.733 | 0.785 |
| BNN | 0.621 | 0.399 | 0.648 | 0.559 | 0.355 | 0.469 | 0.516 |
| BayesAdapter- | 0.273 | 0.273 | 0.273 | 0.251 | 0.248 | 0.309 | 0.275 |
| BayesAdapter | **0.998** | **0.987** | **0.999** | **0.999** | **0.986** | **0.994** | **0.996** |

As for the *MC dropout*, we add dropout-0.3 (0.3 denotes the dropout rate) before the second convolution in the residual blocks in wide-ResNet-28-10, dropout-0.2 after the second and the third convolutions in the bottleneck blocks in ResNet-50, and dropout-0.2 before the last fully connected (FC) layer in MobileNetV2.

For reproducing *Deep Ensemble*, we train 5 *MAP*s separately, and assemble them for prediction and uncertainty quantification. For reproducing *SWAG*, we take use of its official implementation, and leverage 20 MC samples for decision making.

## C  MORE RESULTS FOR UNCERTAINTY ESTIMATION

We provide the comparison on the quality of uncertainty estimates on face recognition in Table 6. It is an immediate observation that *BayeAdapter* outperforms the extensive baselines significantly, and can detect almost all the OOD instances across the validation datasets. By contrast, *BayeAdapter-*, *MAP*, and *BNN* are similarly unsatisfactory. Surprisingly, *MC dropout* exhibits some capacity to detect adversarial instances and DeepFake ones in the face tasks. Comparing these results with those of *MC dropout* on CIFAR-10 and ImageNet, we speculate that such results may stem from the location of deploying dropout in the architecture, which deserves a future investigation.

## D  VISUALIZATION OF SOME OOD DATA

We provide some random samples of the OOD data used for evaluation in Figure 6. Obviously, these samples are pretty realistic and challenging.

## E  VISUALIZATION OF THE LEARNED POSTERIOR

We plot the parameter posterior of the first convolutional kernel in ResNet-50 architecture learned by BayesAdapter on ImageNet. The results are depicted in Figure 7. The learned posterior variance seems to be disordered, unlike the mean. We leave more explanations as future work.

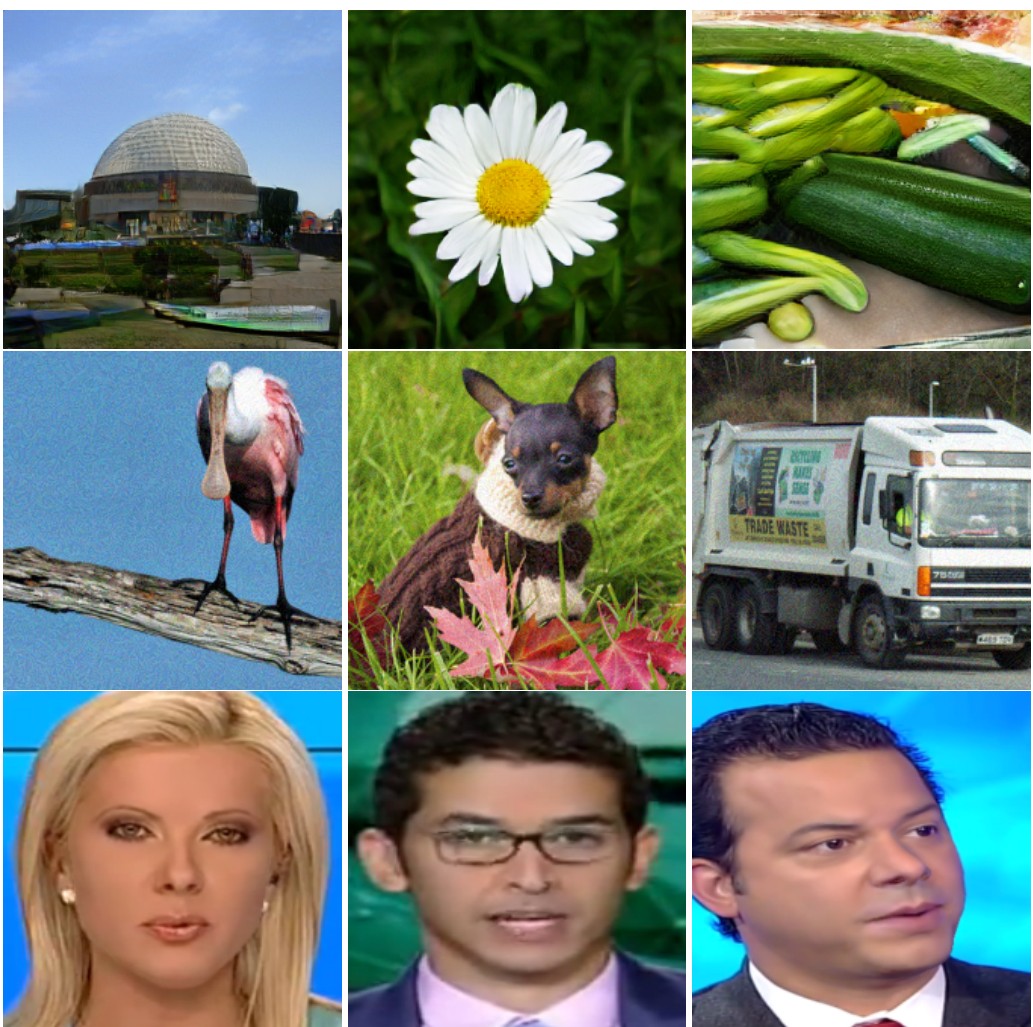

**Figure 6:** Some random samples of the OOD data used for evaluation. The first row refers to the fake samples from BigGAN on ImageNet. The second row refers to the adversarial examples generated by PGD on ImageNet. The third row refers to the fake samples from DeepFake.

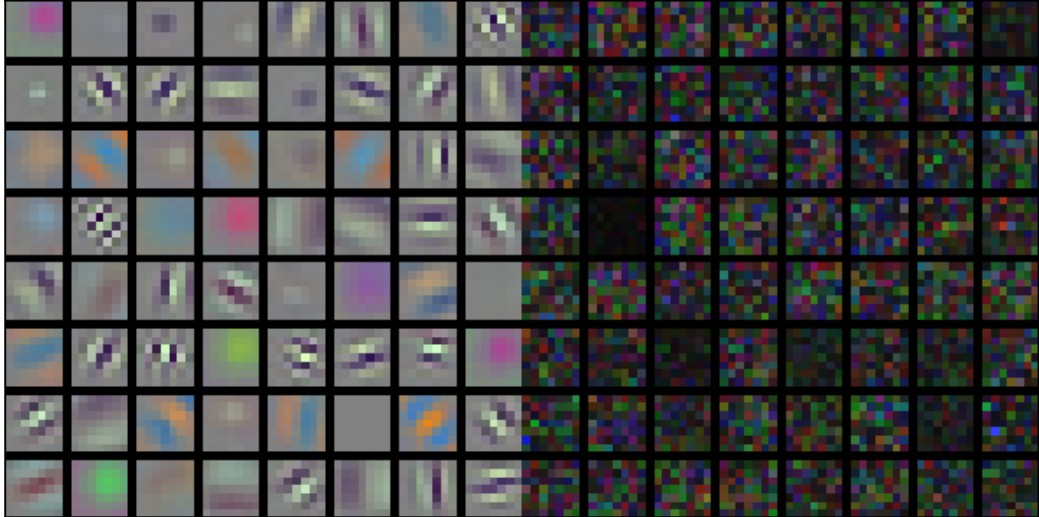

**Figure 7:** Left: the mean of the Gaussian posterior. Right: the variance of the Gaussian posterior. These correspond to a convolutional kernel with 64 output channels and 3 input channels, where every output channel is plotted as a separate image.

