# OpenReview forum: "BayesAdapter: Being Bayesian, Inexpensively and Robustly, via Bayesian Fine-tuning"
_ICLR.cc/2021/Conference — Reject_

### Official Review · AnonReviewer4 · 2020-10-14
**This paper refines existing ideas and places them into a simple framework that produces good results.**

**Rating:** 6
**Confidence:** 3

**Review:**

**Contributions**

This paper proposes a post-hoc approach to obtain model uncertainty estimates from vanilla pre-trained NNs through MFVI fine-tuning. Namely, 1) the authors re-cast the KL divergence in the VI objective as weight decay applied to the variational parameters 2) the authors propose a variance reduction technique for the reparametrisation trick 3) the authors explicitly train their model to produce large model uncertainty on Out of Distribution (OOD) inputs.
Empirically, the proposed methods seems to retain the strong performance, and much of the simplicity, of point-estimate NNs while providing enhanced robustness in terms of uncertainty estimation.


**originality and significance**

To my knowledge, most of the proposed techniques (or variants of them) have appeared before in the literature or are simple extensions of existing approaches: Re-casting MFVI as SGD [Khan et. al., 2018], Decorrelation of reparametrisation gradients across batch elements [Wen et. al., 2018], Training on OOD measurement points to produce large uncertainty [Hendrycks et. al., 2018 and Hafner et. al., 2018] .
However, this work refines these ideas and puts them into a single framework which seems to produce strong results. I view this as a noteworthy contribution which might bring Bayesian Deep Learning closer to real world deployments.

 **clarity**

Most ideas are presented clearly. The paper is well structured and easy to follow. Some passages are slightly ungrammatical but never does this impede the transmission of ideas.

**pros**
* Presents useful practices to make BNNs more mainstream with strong empirical performance.
* Authors provide code for an efficient implementation of exemplar reparametrisation.
* The proposed technique for OOD detection bypasses typical pathologies of MFVI [Ovadia  et. al., 2019] by explicitly optimising variational parameters to produce large model uncertainties OOD.

**cons**
* Exemplar reparametrisation is very similar to Flipout [Wen et. al., 2018]. A comparison of the two would be appreciated.
* The proposed technique for OOD detection is not very principled and has provides no guarantees. It seems empirically successful however.
* The experimental setup is not very clear, even when reading the supplementary sections concerning experimental setup. Some questions I was left with:
	* There are many hyperparameters, how did you find all of them? —Especially the weight decay coefficients.  Are the standard deviations implied by these priors interesting / meaningful in any way?
	* A single Mutual Information threshold is provided. Is this one used for uncertainty calibration training on all tasks? Is it also used when classifying inputs as in-distribution or OOD? If this is the case, it is possible that models without uncertainty calibration training would benefit from using a different threshold. A better metric might be ROC-AUC, as it is threshold agnostic.
* The only baselines provided are other VI approaches. SWAG is known to be a decently strong baseline but it is not evaluated for OOD detection peroformance. The current  state of the art baseline for uncertainty quantification is deep ensembles [Ovadia  et. al., 2019]. These are much more expensive. However, it might be interesting to see how they compare.
* The authors repeat all experiments 3 times but only provide mean results. In some cases, like tables 1 and 2, the values presented are similar across methods. I think that errorbars (standard deviation across 3 runs) would be very informative to the reader.

**Other comments and questions:**
* Typo in title: Bayesian, not Bayeisian
* The maximum predictive entropy (and thus mutual information) will depend on dimensionality of output space (number of classes). In your experimental section, you say you set a single threshold for all models. Could you further comment on this?

**References**
[Wen et. al., 2018]  https://arxiv.org/pdf/1803.04386.pdf
[Khan et. al., 2018] http://proceedings.mlr.press/v80/khan18a/khan18a.pdf
[Ovadia  et. al., 2019] https://papers.nips.cc/paper/9547-can-you-trust-your-models-uncertainty-evaluating-predictive-uncertainty-under-dataset-shift.pdf
[Hendrycks et. al., 2018] https://openreview.net/forum?id=HyxCxhRcY7
[Hafner et. al., 2018] https://arxiv.org/abs/1807.09289

---

> ### Author Response · Authors · 2020-11-14
> **Thank you for the thorough feedback! (Part 2/2)**
>
> #### Q5: Regarding the error bars:
>
> A: We add the variance of the results into the paper. Here is a copy:
>
> | Metric | Acc. (%) | NLL | AP (PGD) | AP (fake) |
> |----------|:-:|---------------|-------------|-------------|
> | CIFAR-10 | 96.82±0.07 | 0.1004±0.0026 | 0.993±0.003 | 0.994±0.001 |
> | ImageNet | top1: 76.26±0.06 top5: 92.96±0.03 | 0.9428±0.0020 | 0.964±0.009 | 0.848±0.037 |
>
> We can see that the results of BayesAdapter exhibit less variance.
>
>
>
> #### Q6: Other questions
>
> A: We have revised the typo.
>
> As we clarified, we only use a shared threshold $\gamma=0.75$ for training across all the settings. And the learned model is robust against the choice of the $\gamma$, testified by an ablation study on $\gamma$ (on CIFAR-10):
>
> | $\gamma$ | 0.25 | 0.50 | 0.75 | 1.0 | 1.50 |
> |-----------|--------|--------|--------|--------|--------|
> | Acc. | 96.93% | 96.70% | 96.82% | 96.74% | 96.79% |
> | AP (PGD) | 0.915 | 0.948 | 0.993 | 0.991 | 0.944 |
> | AP (fake) | 0.910 | 0.981 | 0.994 | 0.994 | 0.988 |
>
> Different dimensionality of output space would result in different scales of uncertainty, and this is also proved by the sub-figure (b) and (d) in Fig. 3. But we can also note that though with different uncertainty scale, most of the normal examples have uncertainty no more than 0.75 (on both CIFAR-10 and ImageNet), thus once we punish the model to assign no less than 0.75 uncertainty for the OOD data, the model naturally acquires the ability to detect OOD data.
>
>
> Per your suggestion, we might achieve better OOD detection by tuning this threshold according to the data and task, though we skipped at the time of submission.

---

> > ### Comment · AnonReviewer4 · 2020-11-21
> > **Thanks for the clarification and standard deviations!**
> >
> > Just a comment on the threshold: Although BayesAdapter might be robust to the choice of threshold, other methods may not. For this reason, ROC-AUC seems like a more reasonable approach for comparison to me -- it considers every possible threshold.

---

> > > ### Author Response · Authors · 2020-11-22
> > > **Thanks for the updated suggestions!**
> > >
> > > We thank you for the new comment. As we clarified in the last part of the answer to Q3, we use the threshold $\gamma=0.75$ only in the training of BayesAdapter (as the other methods do not involve uncertainty regularization). In the evaluation phase, we use the Area Under the Precision-Recall Curve (AP) of the OOD detection to reflect the quality of uncertainty estimates of every method (including BayeApdater and the other methods). It is known that AP is more suitable than ROC-AUC when there is class imbalance. We hope these could relieve the concerns of the reviewer.

---

> ### Author Response · Authors · 2020-11-14
> **Thank you for the thorough feedback! (Part 1/2)**
>
> We thank the reviewer for the constructive suggestions and the acknowledgment of the practical value of our work. We address the detailed concerns in the following.
>
> #### Q1: Comparison to Flipout [Wen et. al., 2018]
>
> A: Thanks for pointing us to a related work. We have cited it in our paper. Though the motivation behind Flipout and the proposed exemplar reparameterization is similar, which is to reduce the variance of the stochastic gradients, the two approaches are implemented differently. Flipout is only suitable for perturbation based MC estimation and is developed upon two assumptions: “(1) the perturbations of different weights are independent, and (2) the perturbation distribution is symmetric around zero”. Obviously, these limitations make Flipout unable to handle complex variational posterior like a FLOW, or an implicit model. Furthermore, as stated in Flipout’s paper, “Flipout’s forward pass requires two matrix multiplications instead of one” and “this incurs the same overhead as the local reparameterization trick [Kingma et al., 2015]”. So Flipout is practically two times slower than the naive stochastic estimation. In contrast, exemplar reparameterization would not introduce extra computational overhead -- with identical FLOPS to the original computation, due to our design in aligning exemplar reparameterization with de facto high-performance computing operators. More importantly, our method places no assumption on the property of the variable to deal with, namely, it is generally applicable to any kind of variational distributions for variance reduction.
>
>
>
> #### Q2: Regarding the theoretical support of the uncertainty regularization
>
> A: Yes, this technique is mostly motivated by the empirical observations that MFVI frequently delivers unreliable uncertainty for some risky OOD data. And it indeed brings successful results. Currently, we mainly focus on developing practical strategies and will leave the development of its theoretical support to future work.
>
>
>
> #### Q3: Hyper-parameters
>
> A: Sorry for some confusing details. The only two important hyper-parameters are the weight decay coefficient $\lambda$ and the uncertainty threshold $\gamma$. Other hyper-parameters for defining PGD or specifying learning rates, etc., all follow standard practice in the DL community. The number of fake data training (1000) and the number of MC samples for evaluation (S) are flexible and not tuned.
>
>
>
> For $\lambda$, we keep it consistent between pre-training and fine-tuning (stated in Algorithm 1), without elaborated tuning, for example, $\lambda=2e-4$ for the wide-ResNet-28-10 architecture on CIFAR-10, $\lambda=1e-4$ for ResNet-50 architecture, and $5e-4$ for MobileNet-V2 architecture. These values correspond to isotropic Gaussian priors with $\sigma_0^2$ as 0.1, 0.0078, and 0.0041 on CIFAR-10, ImageNet, and CASIA, respectively. It is notable that for a “small” dataset like CIFAR-10, a flatter prior is preferred. While on larger datasets with stronger data evidence, we need a sharper prior for regularization.
>
>
> For $\gamma$, we use $\gamma=0.75$ for training across all the scenarios. But it is not used for OOD detection in the testing phase. For estimating the results of OOD detection, we use the non-parametric metric average precision (see the metric part of Section 4), which is the Area Under the Precision-Recall Curve and is more suitable than the mentioned ROC-AUC metric when there is class imbalance.
>
>
>
> #### Q4: More comparison to SWAG and deep ensemble
>
> A: Thanks for the advice. We will reproduce SWAG and provide a more thorough comparison in the final version.
>
>
> On the other side, we emphasize the direct comparison between BayesAdapter (MFVI) and ensemble-based methods is unfair due to the latter’s requirement of orders of magnitude more training efforts. It is no doubt that BayesAdapter is weaker in terms of predictive performance.
>
> If we pursue predictive performance, we can deploy BayesAdapter upon a more performant pre-trained DNN, which has more parameters and better architecture, or is trained with more advanced techniques. But ensemble-based methods struggle to scale up mainly owing to the sheer time and space complexity.

---

> > ### Comment · AnonReviewer4 · 2020-11-21
> > **Thank you for the response!**
> >
> > Thanks for your response.
> >
> > * I would encourage you to place the additional details on hyper parameter selection in the main text or appendix to facilitate reproducing your work.
> >
> > * I agree that it would be unreasonable to expect an MFVI approach to outperform a deep ensemble. Just because your methods would fare worse does not mean the comparison is not useful. Such a comparison would give an idea of how close you can get with a SOTA MFVI approach (yours) compared to the SOTA across all approaches. (This is done in papers that get accepted at top venues: https://papers.nips.cc/paper/2020/file/781877bda0783aac5f1cf765c128b437-Paper.pdf) In other words, it would shed some light on the limitations of MFVI and better help practitioners understand when they should use MFVI and when ensembles are the better option.
> >
> > * You mention *"We take the results of SWAG from its paper due to implementation difficulties."* I have gone through that papers' code before. I agree it is clearly research code quality but it is not impossible to adapt to existing workflows, especially considering the method is quite simple.

---

> > > ### Author Response · Authors · 2020-11-22
> > > **Thanks for the updated suggestions!**
> > >
> > > We appreciate these new suggestions from the reviewer, and we are willing to take them to further strengthen this submission. Currently, we have revised Appendix B to detail the hyper-parameter selection. We will try to reproduce SWAG and deep ensemble and provide a more thorough comparison in the final version.

---

### Official Review · AnonReviewer3 · 2020-10-26
**Providing a simple method to realize potential advantages of Bayesian neural networks**

**Rating:** 6
**Confidence:** 4

**Review:**

This paper proposed one simple and effective way to trainBayesian neural networks (BNN).

Pros:
1. The proposed method is quite simple and cheap to realize, compared to previous Bayesian methods.
2. Extensive experiments on a diverse set of challenging benchmarks have been conducted, which shows several promising results of the proposed method.
3. The proposed idea is novel which distinguishes from most of previous efforts, which try to train BNN from scratch using Bayesian methods. As described in this paper, most of previous methods, though paying much additional efforts than deterministic ones, do not lead to expected results, even with non-diagonal covariance matrices. The BayesAdapter, however, pays little efforts and obtains improvements even with diagonal covariance matrices. From this perspective, this is an encoraging result.

Cons:
1. The results of comparison in Table 1 only repot the average result in 3 runs (3 is kind of small). However, it is better to show the std metric of the result to make the comparison more convincing because the improvement of BayesAdapter in average value is in fact not very apparent, especially compared with MAP. If the variance of the result is large, then there will be large overlap between different methods and thus it is not reasonable to claim that there is an apparent advantage over previous methods.

2. In evaluating the result of BayesAdapter,  MC samples are used. What if only using the mean value of the posterior?  Compared to deterministic methods like MAP, inference using MC is more costly.   In addition, it is suggested to provide some visualizations of the posterior distribution after BayesAdapter.

3. Based on results in Table 3, BayesAdapter- performs similar as baselines, which indicates that the improvement comes from calibrating the uncertainty estimation. This leads to another question: what if we also use such calibration for the baseline methods. It would be interesting to make such a comparison.

---

> ### Author Response · Authors · 2020-11-14
> **Thank you for the thorough feedback!**
>
> We thank the reviewer for the positive review. In this following, we address the detailed comments.
>
>
>
> #### Q1: The std metric
>
> Sorry for missing the std. Here we provide it on CIFAR-10 and ImageNet benchmarks.
>
> | Metric | Acc. (%) | NLL | AP (PGD) | AP (fake) |
> |----------|:-:|---------------|-------------|-------------|
> | CIFAR-10 | 96.82±0.07 | 0.1004±0.0026 | 0.993±0.003 | 0.994±0.001 |
> | ImageNet | top1: 76.26±0.06 top5: 92.96±0.03 | 0.9428±0.0020 | 0.964±0.009 | 0.848±0.037 |
>
> We can see that the results of BayesAdapter are relatively stable, outperforming BNN and MAP with statistical evidence on ImageNet.
>
>
>
> Following a similar suggestion by R1, we have revised the paper to make clear the major goal of this work, which is to quickly and cheaply adapt a pre-trained DNN to be Bayesian without compromising performance when facing new tasks, instead of delivering a mechanism for learning better BNNs.
>
>
>
> #### Q2: MC samples and posterior appearance
>
> A: We kindly point out that the results of deterministic inference with only the posterior mean of BayesAdapter are provided in the ablation study “The impacts of ensemble number” and Figure 4. It is clear that with more than around 20 MC samples, Bayes ensemble (the green line) can achieve better prediction results than the deterministic inference (the yellow line). This reflects that the learned posterior does not suffer too much from mode collapse, which is popularly witnessed on mean-field variational inference by the community.
>
>
>
> As suggested, we plot the parameter posterior of the first convolutional kernel in ResNet-50 architecture learned by BayesAdapter on ImageNet. The results are depicted in Appendix E. The learned variance seems to be disordered, unlike the mean. We leave more explanations as future work.
>
>
>
> #### Q3: Apply uncertainty regularization to other models
>
> A: Yes, we totally agree with this point. We conducted such an experiment, and observed improved uncertainty estimation in the initial phase when applying this technique to other BNN methods, including the BNN baseline. These further confirm the effectiveness and universality of the proposed uncertainty regularization technique. We will try to add complete results in the final version since that training BNNs from scratch is time-consuming.

---

> > ### Comment · AnonReviewer3 · 2020-11-24
> > **Thanks for the response**
> >
> > Thank you for the authors response to my concerns.
> >
> > Overall, I have a similar feeling as R2.  It provides an interesting and simple way to obtain a Bayesian-like neural network, but still no fundamental improvement that brings excitement to the community. Compared to previous issues, it nonetheless provides a new perspective. Thus I would like to keep my previous score.

---

### Official Review · AnonReviewer2 · 2020-10-29
**Some interesting ideas; limited novelty; experimental section could be improved.**

**Rating:** 5
**Confidence:** 4

**Review:**

The paper explores the variational training of Bayesian neural networks. It proposes to improve the quality of the inferred variational posterior and computational efficiency of the procedure by (i) better initialization (mean parameters are initialized at the MAP) (ii) reducing variance in the Monte Carlo approximated evidence lower bound by increasing the number of weight samples (one per datapoint in a batch) (iii) a posterior regularization encouraging higher uncertainty on adversarially generated or other “near OOD” data.

The authors use the term BayesAdapter to refer to the process of running black-box variational inference from a fully factorized variational approximation with mean initialized at the MAP estimate and randomly initialized variances. The fact that variational inference (especially those employing fully factorized approximations) are susceptible to poor local optima and that better initializations can help navigate these local optima is widely known. The fact that better initializations can lead to somewhat improved posterior approximations is not surprising. Such initializations are also standard practice when employing stochastic gradient MCMC techniques and Laplace approximations (where it is a requirement). Reducing variance by increasing the number of weight samples to one per datapoint in a batch is another straightforward idea, and it is unclear whether it can be claimed as a contribution of the current paper. Kingma et al., in their local re-parameterization considered a variant with per data samples as well. The uncertainty regularization is indeed novel and appears effective (but the experiments illustrating its benefits need to be better explained).

Given the modest methods contributions, the empirical section needs to be particularly strong to demonstrate that the combination of these incremental improvements provides meaningful empirical advantages. To their credit, the authors demonstrate their approach on several large datasets and do provide experiments for vetting different aspects of the proposed extensions to variational BNN training. However, many experiments are missing details and some are lacking key comparisons. Overall this section could be significantly strengthened.

* Tables 1 and 2 need to include comparisons against deep ensembles and multi-SWAG (https://arxiv.org/pdf/2002.08791.pdf). If the goal of this paper is to claim that variationally trained BNNs (with the proposed improvements) are useful in practice, a natural question to ask is whether they are competitive with far simpler ensembling approaches that are able to account for the multimodality of the posterior surface, unlike variational BNNs.
* How was the calibration threshold $\gamma$ chosen for these experiments? How sensitive is the performance to this choice? Ideally, the authors would include results with different settings of $\gamma$. How were the prior precisions selected (which determine $\lambda$ set?  My main worry is that the marginal improvements provided by Bayesadapter variants over BNNs disappear when making slightly different parameter choices. It would be great if the authors can demonstrate that this isn’t the case.
* Section 4.2 needs more details about the experimental setup. Did the 1000 / 10000 OOD training/test examples include both images created via PGD and SNGAN (for CIFAR 10) and PGD and BigGAN (for imagenet)? If so, how many from each source? If not, it would be interesting to see cross performance — using PGD images for training and SNGAN images for testing.
* In Table 4, it doesn’t make sense to include ECE numbers from a different architecture trained via SWAG. These numbers are not comparable. Also, interestingly, both BayesAdapter variants have lower ECE scores than vanilla BNN on CIFAR, suggesting poorer calibration. Do the authors have an explanation for this?

Based on concerns about both novelty and experiments I am currently leaning towards a reject, but could be convinced otherwise based on the authors’ response and additional comparisons.

---

> ### Author Response · Authors · 2020-11-14
> **Thank you for the thorough feedback! (Part 2/2)**
>
>
>
> #### Q4: Regarding setup
>
> We are sorry for causing the misunderstanding. To clarify:
>
> For the training, we use only 1000 fake examples (e.g., those from SNGAN on CIFAR-10 and from BigGAN on ImageNet) and all the _uniformly perturbed training examples_ for optimizing uncertainty regularization (see the last part of Section 3). We did not include PGD perturbed examples into training because that resembles adversarial training and is time-consuming. For evaluation, we estimate the uncertainty on a held-out set of fake examples and _PGD perturbed validation examples_ and report the results.
>
>
>
> We want to point out that the idea to “see cross performance” might not be helpful --- with the uncertainty estimation trained on adversarially perturbed data, we cannot expect it to successfully generalize to the examples produced by GAN which have significantly different fingerprints (see Figure 6 in Appendix).
>
>
>
> We have offered a study on if the learned uncertainty could reasonably generalize in part 2 of Section 4.3. The results give a positive answer.
>
>
>
> #### Q5: Regarding ECE
>
> Thanks for the advice. We clarify that on CIFAR-10, SWAG also uses wide-ResNet-28-10, and shows weaker calibration than BayesAdapter with a substantial margin. The comparison to SWAG with a different architecture on ImageNet may indeed be less meaningful, and we have revised this.
>
> Regarding the second point, we emphasize the core notion of Bayesian deep learning: the predictive confidence is usually unreliable, thus we need a better measure like predictive uncertainty. With the superiority of the uncertainty estimation of BayesAdapter validated by Table 3 and Table 5, its weaker ECE does not substantially undermine its practical value. As shown in the ablation study “Uncertainty-based rejective decision” and Figure 5, we can leverage the predictive uncertainty to achieve robust rejective decision making instead of using predictive confidence. Anyway, returning to this phenomenon, we speculate this is because the fine-tuning start point _MAP_ has too bad ECE and the fine-tuning rounds are few.

---

> ### Author Response · Authors · 2020-11-14
> **Thank you for the thorough feedback! (Part 1/2)**
>
> We thank the reviewer for the effort in assessing our work and the constructive comments. First, we clarify the contribution of our work, and then we answer the questions in detail.
>
>
>
> #### Q1: Regarding the contributions
>
> We thank the reviewer for the thorough literature review and for relating our work with extensive theoretical and empirical support. Here, we want to clarify that the proposed BayesAdapter is not a naive combination of the three aspects mentioned by the reviewer. The central goal across these learning strategies is to constitute a practically useful tool to “bring Bayesian Deep Learning closer to real-world deployments” at a low cost, as appreciated by the other reviewers.
>
> Technically, the introduction of the pre-training & fine-tuning framework, which is especially popular in the deep learning community recently, into the learning of variational BNNs is novel, providing us with the opportunities to achieve fast and cheap Bayesian inference when facing new tasks.
>
> Regarding the exemplar reparameterization for variance reduction, though the idea is not new, we develop an insightful implementation (see Fig. 2) to make the computations practically approachable and highly compatible with high-performance computation kernels. We emphasize that our implementation is distribution agnostic, implying that it is generally applicable to any forms of variational distribution for reducing gradient variance, unlike local reparameterization [Kingma et al., 2015] which is typically limited to exponential family, in particular, Gaussian distribution.
>
> As approved by the reviewer, the uncertainty regularization is well-motivated and creative. We have offered a wide range of experiments to demonstrate its effectiveness. As shown in Table 3 and Table 5 (in Appendix), the uncertainty regularization aids to endow the BNNs with near-perfect ability to perceive adversarial and fake examples. Besides, Fig. 3 provides a direct illustration of how this term rectifies the uncertainty quantification of the BNN.
>
>
>
> #### Q2: Comparison to deep ensemble and multi-SWAG
>
> Thank you for the advice. However, we need to clarify that it is not necessary to show the superiority of BayesAdapter over ensemble-based methods to confirm its practical value. In fact, BayesAdapter considers a more common and realistic case: in some real-world tasks, we have obtained a trained DNN and we expect to equip the DNN with uncertainty estimation to make the decision-making benefit from Bayes principle (i.e., leverage uncertainty to reject to predict for uncertain data). BayesAdapter is naturally suitable for this situation by adapting the DNN to be BNN with non-degraded performance (proved by the experiments) at a low expense. This is what ensemble-based methods cannot do.
>
> The direct comparison between BayesAdapter (MFVI) and ensemble-based methods is obviously unfair due to the latter’s requirement of orders of magnitude more training efforts. It is no doubt that BayesAdapter is weaker in terms of predictive performance.
>
> If we pursue predictive performance, we can deploy BayesAdapter upon a more performant pre-trained DNN, which has more parameters and better architecture, or is trained with more advanced techniques. But ensemble-based methods struggle to scale up extremely owing to the sheer time and space complexity.
>
>
>
> At last, we emphasize that the compared SWAG is already a decently strong baseline, as mentioned by R4.
>
>
>
> #### Q3: The hyper-parameter setting
>
> A: We empirically observed that the normal examples usually present <0.75 mutual information uncertainty, across the studied scenarios. Then we use $\gamma=0.75$ in the regularization to push both the adversarial and fake data to exhibit uncertainty no less than 0.75.
>
>
>
> As suggested by the reviewer, we also perform an ablation study regarding $\gamma$ on CIFAR-10:
>
>
> | $\gamma$ | 0.25 | 0.50 | 0.75 | 1.0 | 1.50 |
> |:-: | :-: | :-: | :-: | :-:|:-:|
> | Acc. | 96.93% | 96.70% | 96.82% | 96.74% | 96.79%|
> | AP (PGD) | 0.915 | 0.948 | 0.993 | 0.991 |0.944|
> | AP (fake) | 0.910 | 0.981 | 0.994 | 0.994 |0.988|
>
> The results reveal that values of $\gamma \in [0.75, 1.0]$ may be good choices for OOD detection.
>
>
>
> Note that we keep the weight decay coefficient $\lambda$ consistent between pre-training and fine-tuning (stated in Algorithm 1), following the common practice in DL, without elaborated tuning, for examples, $\lambda=2e-4$ for the wide-ResNet-28-10 architecture on CIFAR-10, $\lambda=1e-4$ for ResNet-50 architecture, and $5e-4$ for MobileNet-V2 architecture.

---

### Official Review · AnonReviewer1 · 2020-10-29

**Rating:** 6
**Confidence:** 4

**Review:**

This paper introduces a fast way to get Bayesian posterior by using a pretrained deterministic model. Specifically, the authors first train a standard DNN model and then use it to initialize the variational parameters. Finally the variational parameters are optimized through standard variational inference (VI) training. To further improve uncertainty estimate, the authors propose an uncertainty regularization which maximizes the prediction inconsistency on out-of-distribution (OOD) data. Experiments including image classification and uncertainty estimates are conducted to demonstrate the proposed method.

The idea of this paper is quite simple: initialize the mean in variational parameters by a pretrained DNN. Thus the method is cheap and simple enough to use broadly in practice. The authors did reasonable empirical tests. I especially appreciate the ablation study which helps understand the method a lot. The paper is well-written and easy to follow.

I mainly have the following concerns about the paper.

- One of the main motivations to use a pretrained DNN is that BNN learned from scratch is worse than its corresponding DNN. I feel this claim is misleading. Many papers have shown that BNNs trained from scratch outperform DNNs. Particularly the paper [Wenzel et.al. 2020] which the authors cited to support their claim clearly shows that BNN (with reasonable temperature) is significantly better than DNN in predictive performance. But I do agree that the proposed method is cheaper than training BNNs from scratch.

- The experimental results verify the effectiveness of the proposed method. But the results of the proposed method seem to be worse than BNNs training from scratch (e.g. the ImageNet results in [Cyclical Stochastic Gradient MCMC for Bayesian Deep Learning, ICLR 2020] are much better). This also supports my first point, that the main benefit of the proposed method is to get the Bayesian posterior fast and cheaply, instead of to improve BNNs’ performance. I think the authors should revise the claim to be more precise.

- The proposed method is closely related to [Specifying Weight Priors in Bayesian Deep Neural Networks with Empirical Bayes, AAAI 2020] which also uses a pretrained DNN as the initialization of variational parameters. How does the proposed method compare to it theoretically and empirically? The main idea seems quite similar; the only difference is that the proposed method does not use the pretrained model as prior in VI. Due to the similarity, I think a comparison is necessary.

- The authors argue that BNN training suffers from suboptimal local optima. Could the authors provide evidence/citations to support this claim? I do not think it is true. Perhaps it is true only for a few BNN methods such as BNN using naive VI.

- As the proposed method is essentially a VI method, it would be interesting to see comparisons to SOTA VI methods.

Overall I'm positive about this paper and would be happy to increase my score if my concerns are addressed.

---

> ### Author Response · Authors · 2020-11-14
> **Thank you for the thorough feedback!**
>
> We thank the reviewer for the positive feedback. We are encouraged by the acknowledgment of the practicability of the proposed approach and the thoroughness of the experiments. We address specific comments below and have updated the paper accordingly.
>
> #### Q1: Concern on “BNN learned from scratch is worse than its corresponding DNN”.
>
> To clarify, “BNNs learned from scratch” here refer to those without explicitly sharpened posterior (i.e., cold posterior). As evidenced in [Wenzel et. al 2020], they typically demonstrate worse performance than the corresponding DNNs. Sharpening the posterior w.r.t. some validation metrics might improve performance but will cause the learned posterior to aggressively deviate from the Bayesian paradigm [Wenzel et al., 2020], possibly compromising the major benefits of BNNs such as calibrated uncertainty estimation.
>
>
>
> #### Q2: Concern on “improved performance”
>
> We agree with the reviewer’s point, and make more clarifications: the improved performance we claimed corresponds to the comparison between our approach and baselines trained from scratch, e.g., between BayesAdapter/BayesAdapter- and BNN on ImageNet classification (Table 1) and face recognition (Table 2).
>
> Instead of delivering a new mechanism for learning better BNNs, the major goal of this work is to quickly and cheaply adapt a pre-trained DNN to be Bayesian without compromising performance when facing new tasks. Such a goal is practical and useful as it enables one to first employ advanced techniques to train a DNN with strong performance and then cheaply adapt it to be a BNN.
>
> We have revised the paper to make the claim more appropriate.
>
>
>
> #### Q3: Comparison to [Krishnan, 2020]
>
> BayesAdapter connects to [Krishnan, 2020] in that the variational configurations of BayesAdapter and [Krishnan, 2020] are both based on MAP. With the prior specified as MAP mean and unit variance, the primary objectives of [Krishnan, 2020] are also to speed up the learning and to bypass the potential local optima of the posterior (see Fig. 1 of [Krishnan, 2020]). Yet, beyond these, BayesAdapter is further designed to achieve good user-friendliness, improved learning stability, and trustable uncertainty estimation, by virtue of optimizers with built-in weight decay, exemplar reparameterization, and uncertainty regularization, respectively. These designs significantly boost the practicability of the proposed method, especially in real-world/large-scale settings.
>
> We’ve added these discussions and comparisons to the revision (see related work section).
>
>
>
> #### Q4: Regarding “BNN trained from scratch suffers from suboptimal local optima”
>
> Yes, for mean-field variational BNNs, this claim is supported by the extensive results in [Krishnan, 2020] and the comparisons between BayesAdapter/BayesAdapter- and BNN in our experiment section.
>
> As mentioned by R2, stochastic gradient MCMC also benefits a lot from a good initialization. For variational BNNs with more complicated posterior, this claim is not rigorous owing to the mismatch between the posterior (e.g., a FLOW) and the pre-trained parameters. We have revised this claim.
>
>
>
> #### Q5: Comparisons to SoTA VI methods
>
> Thanks for the advice. As stated by R4, the SWAG baseline we compared to is a decently strong baseline. We tried some advanced VI methods like VOGN and noisy-KFAC, but encountered difficulties to scale them up to models with more parameters (e.g., wide-ResNet-28-10) and large datasets (e.g., ImageNet), or compatibility issues with practical data augmentation and batch normalization techniques. Nevertheless, we will try to reproduce some of them and include the results in the final version.
>
>
>
> [1] *Specifying Weight Priors in Bayesian Deep Neural Networks with Empirical Bayes, Krishnan et al., AAAI 2020*
>
> [2] *How Good is the Bayes Posterior in Deep Neural Networks Really? Wenzel et al., ICML 2020*

---

> > ### Comment · AnonReviewer1 · 2020-11-24
> > **Thank you for your responses**
> >
> > After reading the responses, I have the following concerns.
> >
> > - The differences between the proposed method and [Krishnan, 2020] seem to be using more weight samples and uncertainty regularization on out-of-distribution (OOD) data. Since [Krishnan, 2020] also uses MAP as initialization in VI, and shows several empirical results including predictive performance, uncertainty estimate and OOD detection. The novelty of using MAP as initialization in VI in the paper is decreased. I think you should rephrase the contributions of the paper, and mainly show the benefits of using more weight samples and uncertainty regularization techniques in the experiments. Also, I think it is necessary to include a comparison with [Krishnan, 2020].
> >
> > - Using [Wenzel et.al], which is an MCMC-based method, to motivate the proposed method seems incorrect. Since the method is a VI-based BNN, you should instead provide evidence that MFVI-based BNN learned from scratch is bad.
> >
> > - Since the proposed method is a variant of VI methods, I still expect a comparison with SoTA VI methods, even if on small datasets.

---

> > > ### Author Response · Authors · 2020-11-25
> > > **Further clarification**
> > >
> > > We thank you for your feedback. We will revise the paper accordingly, and we expect to provide further clarification to alleviate the reviewer's concerns:
> > >
> > >
> > > - We accept the suggestion of rephrasing the contributions of the paper. But we expect to emphasize that the goal of [Krishnan, 2020] is to solve the problem of prior specification problem of variational BNNs. There is no particular design in the consideration of practicability. However, BayesAdapter leverages the pre-training & fine-tuning workflow to significantly boost the efficacy and scalability of VI based BNNs (e.g., deliver an outperforming variational BNN on *ImageNet*), which is of central importance for BNNs' real-world adoption.
> > >
> > > - [Wenzel et.al] particularly list a range of  SoTA variational BNNs in Sec 2.3 of their paper to draw the conclusion that "the cold posterior problem has left a trail in the literature, and in fact we are not aware of any published work demonstrating well-performing Bayesian deep learning at temperature T = 1". So we refer to [Wenzel et.al] to evidence that MFVI-based BNN learned from scratch is bad if without cold posterior.
> > >
> > > - We will provide the results of competitive VOGN or NEK-FAC in the final version.

---

> > > > ### Comment · AnonReviewer1 · 2020-11-25
> > > > **Response**
> > > >
> > > > I do not see how you draw the conclusion "MFVI-based BNN _learned from scratch_ is bad" from "the cold posterior problem has left a trail in the literature, and in fact we are not aware of any published work demonstrating well-performing Bayesian deep learning at temperature T = 1". Particularly, Figure 13 in [Wenzel et.al.] shows that initializing MCMC with the MAP solution does not solve the cold posterior problem.

---

> > > > > ### Author Response · Authors · 2020-11-25
> > > > > **Response**
> > > > >
> > > > > Thank you again for the immediate reply!
> > > > >
> > > > > -  We kindly point out that almost all the existing variational BNNs perform *from-scratch* learning (e.g., the noisy KFAC, noisy EKFAC, and VOGN mentioned in [Wenzel et.al.]). They typically cannot demonstrate good performance without cold posterior, thus we conclude that MFVI-based BNN *learned from scratch* is typically worse than its DNN counterpart when "cold posterior" strategies are not applied (as suggested by the reviewer, we have revised our claim in the submission).
> > > > >
> > > > > - Due to the high alignment between the pre-trained DNNs and the variational BNNs to learn, we can easily guarantee non-degraded performance during Bayesian fine-tuning even without sharpened posterior, verified by our experiments.

---

### Author Response · Authors · 2020-11-14
**Paper update overview**


We thank all the reviewers for their careful reading and constructive feedback. In the revised version, we addressed the comments to strengthen our paper. In summary, here are the main changes that we made to the paper:
- Revise the claim on BNNs trained from scratch.
- Revise the claim on improved performance of BayesAdapter.
- Add the related work [Krishnan, 2020].
- Clarify the setting of $\gamma$ and provide an ablation study on it.
- Add std for the results of BayesAdapter.
- Revise the claim of contributions.
- Cite Flipout.
- Revise the typos.

We are open to more suggestions.

---

### Author Response · Authors · 2020-11-23
**Looking forward to further feedback!**

Dear Reviewers,

Thank you again for your valuable comments and suggestions, which are really helpful for us. We have uploaded new revisions and posted responses to the proposed concerns and questions. We have reproduced deep ensemble on CIFAR-10 benchmark, here are the classification results:

| Method | Acc. (%) | NLL  |
|----------|:-:|--------------|
| Deep ensemble | 97.40 | 0.0869  |
| BayesAdapter- |97.09 | 0.0945  |
| BayesAdapter |96.82±0.07 | 0.1004±0.0026  |

We have also included deep ensemble and SWAG into the OOD detection on CIFAR-10 to assess the quality of their uncertainty estimates. Here are the results:

| Method | AP (PGD) | AP (fake) |
|----------|:-:|--------------|
| Deep ensemble | 0.427 | 0.812  |
| SWAG |0.316 | 0.816  |
| BayesAdapter |0.993±0.003 | 0.994±0.001  |

Though BayesAdapter is defeated by deep ensemble in the aspect of classification performance, it can provide significantly more reliable uncertainty estimates than deep ensemble and SWAG for detecting challenging OOD data. We hope these results help to relieve the concerns of the reviewers.

At last, we deeply appreciate it if the reviewers can take some time to return further feedback on whether our responses and extra experiment results solve their concerns. If there is any other question, we will try our best to provide satisfactory answers.

Best,
The authors

---

### Decision · Program_Chairs · 2021-01-07
**Final Decision**

**Decision:**

Reject

**Comment:**

This paper aims at improving the adoption of Bayesian NNs by providing a practical and user friendly variational inference method. The main ideas consist of two parts:
1. Warm-start the variational inference from a pre-trained deterministic NN. It takes advantage of existing deep learning library features for easy implementation including weight decay, batch matrix multiplication, etc.
2. Calibrating uncertainty estimation for out-of-domain detection using adversarial examples.

Pros:
1. A practical way of implementing DNN variational inference with reduced variance, without sacrificing classification accuracy of the pretrained NN model.
2. Significantly better OOD detection accuracy compared to other BNN approaches without taking OOD into account explicitly.

Cons:
1. During discussion, it becomes clear that most of the techniques have been proposed similarly in the literature. Krishnan, 2020 applied BNN starting from MAP of NN, Flipout (Wen et. al., 2018) applies instance-wise sampling, Hendrycks et. al., 2018 and Hafner et. al., 2018 improves detection accuracy by training on OOD examples. The novelty of the proposed method is therefore limited.
2. There's not much benefit on the classification performance compared to the initial MAP and is inferior to MCMC-based SOTA BNNs. One of the reviewers considers the SGLD-type approach may be more appealing to ML practitioners with the overhead of VI in training additional variance parameters.
3. The authors argue MCMC-based BNN methods cannot achieve good performance without temperature scaling. But the main performance improvement of the paper is in the OOD detection with uncertainty regularization that modifies the posterior as well. The method of training with OOD samples is orthogonal to applying Bayesian inference to NNs, and the detection performance is limited to the distribution close to examples during training.

This paper falls on the borderline for acceptance. With the goal of improving adoption of BNN in practice, it is not convincing yet making mean field VI easier to implement could realize it without achieving competitive performance.